# Genome-Wide Analysis of *OPR* Family Genes in Cotton Identified a Role for *GhOPR9* in *Verticillium dahliae* Resistance

**DOI:** 10.3390/genes11101134

**Published:** 2020-09-27

**Authors:** Shichao Liu, Ruibin Sun, Xiaojian Zhang, Zili Feng, Feng Wei, Lihong Zhao, Yalin Zhang, Longfu Zhu, Hongjie Feng, Heqin Zhu

**Affiliations:** 1State Key Laboratory of Cotton Biology, Institute of Cotton Research of Chinese Academy of Agricultural Sciences, Anyang 455000, China; liushichao29@163.com (S.L.); sunruibin@caas.cn (R.S.); fengzili@caas.cn (Z.F.); weifeng0108@163.com (F.W.); zhaolihongqq@163.com (L.Z.); yalinzhang2012@163.com (Y.Z.); 2College of Plant Science and Technology, Huazhong Agricultural University, Wuhan 430070, China; lfzhu@mail.hzau.edu.cn; 3Zhengzhou Research Base, State Key Laboratory of Cotton Biology, School of Agricultural Sciences, Zhengzhou University, Zhengzhou 450001, China; xiaojianzhang993@163.com

**Keywords:** OPR, cotton, gene family, *Verticillium dahliae*, VIGS

## Abstract

The 12-oxo-phytodienoic acid reductases (*OPRs*) have been proven to play a major role in plant development and growth. Although the classification and functions of *OPRs* have been well understood in Arabidopsis, tomato, rice, maize, and wheat, the information of *OPR* genes in cotton genome and their responses to biotic and abiotic stresses have not been reported. In this study, we found 10 and 9 *OPR* genes in *Gossypium hirsutum* and *Gossypium barbadense*, respectively. They were classified into three groups, based on the similar gene structure and conserved protein motifs. These *OPR* genes just located on chromosome 01, chromosome 05, and chromosome 06. In addition, the whole genome duplication (WGD) or segmental duplication events contributed to the evolution of the *OPR* gene family. The analyses of cis-acting regulatory elements of *GhOPRs* showed that the functions of *OPR* genes in cotton might be related to growth, development, hormone, and stresses. Expression patterns showed that *GhOPRs* were upregulated under salt treatment and repressed by polyethylene glycol 6000 (PEG6000). The expression patterns of *GhOPRs* were different in leaf, root, and stem under *V. dahliae* infection. *GhOPR9* showed a higher expression level than other *OPR* genes in cotton root. The virus-induced gene silencing (VIGS) analysis suggested that knockdown of *GhOPR9* could increase the susceptibility of cotton to *V. dahliae* infection. Furthermore, *GhOPR9* also modulated the expressions of jasmonic acid (JA) pathway-regulated genes under the *V. dahliae* infection. Overall, our results provided the evolution and potential functions of the *OPR* genes in cotton. These findings suggested that *GhOPR9* might play an important role in cotton resistance to *V. dahliae*.

## 1. Introduction

In plants, jasmonic acid (JA) is a lipid-derived signaling phytohormone and activates defense against various stresses, such as insects, pathogen, salt, temperature, and wounding [1,2]. Besides, it also contributes to modulating plant growth and development, such as seed germination, tendril coiling, pollen maturation, fruit ripening, and root growth [3,4]. Furthermore, the JA signal pathway combines with the salicylic acid (SA) signal pathway, ethylene (ET) signal pathway, and abscisic acid (ABA) signal pathway to form a complex network [5,6,7].

Jasmonic acid biosynthesis is originated from α-linolenic, followed by the function of lipoxygenase (LOX), allene oxide synthase (AOS), and allene oxide cyclase (AOC) to form 12-oxophytodienoic acid (OPDA) via the octadecanoic pathway [8,9,10]. *OPRs* are the key enzyme to catalyze the conversion from OPDA to 12-oxo-phytodienoic acid (OPC-8:0), a reaction that is a key process of JA biosynthesis [11,12]. *OPR* genes belong to the old yellow enzyme (OYE) family. They are categorized as flavin mononucleotide (FMN)-dependent oxidoreductases. The Pfam analysis revealed that *Oxidored_FMN* (ID:PF00724) was the specific domain in OPR proteins. The OPR was first purified from cell cultures of *Corydalis sempervirens*, and the first *OPR* homologous gene in a plant was cloned from *Arabidopsis thaliana* [13]. There were three *OPR* genes in *A. thaliana*: *AtOPR1*, *AtOPR2*, and *AtOPR3*. They were classified into two groups—group I (OPRI) and group II (OPRII)—based on their different substrates. *AtOPR1* and *AtOPR2* were classified into group I, catalyzed 9R,13R-OPDA, which is not related to JA biosynthesis. The *AtOPR3* (OPRII) preferentially catalyzed 9S,13S-OPDA, an intermediate precursor of JA biosynthesis. A previous study showed that AtOPR3 was the only enzyme of OPR to reduce the correct stereoisomer of OPDA to produce JA required for male gametophyte development [14].

Subsequently, *OPR* genes were identified in dicots and monocots, such as pea [15], maize [16], tomato [17], tea [18], rice [19], and wheat [20]. In contrast to dicots, the *OPR* genes were classified into five subgroups in monocots [21]. In wheat, 48 *OPR* genes were classified into five groups: there were six in Group I, four in Group II, 33 in Group III, three in Group IV, and two in Group V [20]. The *OPR* genes expression levels were different in different tissues of plants or under various biotic and abiotic stresses, based on their distinct biological functions. In maize, eight *OPR* genes were identified. *ZmOPR1* and *ZmOPR2* were transiently induced by SA, chitooligosaccharides, and by infection with pathogens, but not by wounding. However, *ZmOPR6*, *ZmOPR7*, and *ZmOPR8* were highly induced by wounding or JA, not involved in response to pathogens infection and SA [16]. In wheat, transgenic plants of *TaOPR1* could improve the tolerance to salinity in an ABA-dependent manner [22]; the expression of TaOPR2 can be induced by wounding, drought, MeJA, *Puccinia striiformis f. sp. Tritici*, and *Puccinia recondite f. sp. Tritici* [23]. Overexpression of *AtOPR3* in hexaploid wheat could enhance the wheat freezing tolerance [24]. Under the drought stress in rice, *OsOPR7* expression level was upregulated to a peak within 0.5 h [25]. In *Solanum lycopersicum*, the *SiOPR3* plants in which the *SlOPR3* was knocked down accumulated less OPDA and JA-lle under *Botrytis cinerea* infection than the control and had increased susceptibility to *B. cinereal* [26]. In upland cotton (*Gossypium hirsutum*), the function of GhOPR3, which was 76% homologous with the amino acid sequence of AtOPR3, has been well characterized. GhOPR3 can be phosphorylated by GhCPK33 at threonine-246 in peroxisomes. Thereby, the protein of GhOPR3 was decreased, which consequently suppressed JA biosynthesis and reduced the resistance of cotton to *V. dahliae* [27].

Cotton is an important economic and natural fiber crop worldwide. Its growth and yield are restricted by various biotic and abiotic stresses in the field. Verticillium wilt, which is caused by the soilborne fungus *V. dahliae*, is a devastating vascular disease in cotton. There are no upland cotton cultivars displaying high resistance to *V. dahliae* [28]. Nevertheless, the genome-wide identification of *OPR* genes in *Gossypium* has not been explicitly studied. In this study, *OPRs* genes were identified by the method of genome-wide analysis in *Gossypium*. Then, the putative 10 *OPR* genes of *G. hirsutum*, nine *OPR* genes of *G. barbadense*, five *OPR* genes of *Gossypium arboreum*, and three *OPR* genes of *Gossypium raimondii* were used to construct the gene phylogenetic tree. Ten *GhOPRs* and nine *GbOPRs* were analyzed, and gene structure, protein conserved motif, chromosome localization, and cis-acting regulatory elements were predicted in *G. hirsutum*. The results indicated that the *OPR* genes have potential functional and they were evolutionary in cotton. Subsequently, expression levels of *GhOPRs* under PEG6000 (two tissues and four time points), NaCl (two tissues and four time points), and *V. dahliae* (three tissues and eight time points) stresses were determined using RT-qPCR to better understand the *OPR* functions in stress responses. Silencing *GhOPR9* in *G. hirsutum* can compromise cotton resistance to *V. dahliae*. Furthermore, we also found that *GhOPR9* can modulate the expression levels of JA biosynthesis genes under *V. dahliae* infection. These results could contribute to screening more potential functional genes in order to improve resistance against biotic and abiotic stresses in cotton.

## 2. Materials and Methods

### 2.1. Identification of OPR Family Genes in Cotton

Proteomes and gene annotation data of *G. hirsutum* and *G. barbadense* are available from the newly assembled genome [29]. To identify *OPR* genes in cotton, the Hidden Markov Model (HMM) models of *Oxidored_FMN* (PF00724) were download from the Pfam database (http://pfam.xfam.org/) [30] and were used as query to conduct a homologous search (E-value < 1) against proteomes of *G. hirsutum* and *G. barbadense* by HMMER v.3.1b2 software [31], respectively. Besides, protein sequences of *A. thaliana OPRs* were retrieved from The Arabidopsis Information Resource (TAIR) database (https://www.arabidopsis.org/) and served as queries to perform similarity search (E-value < 1 × 10^−5^, identity > 50%) against *G. hirsutum* and *G. barbadense* proteomes using BLAST + v.2.6.0, respectively [32]. The sum total of object items in HMMER and BLAST search results were subjected to further filtering by InterProScan v.5.32-71.0. Items [33] containing the characteristic *Oxidored_FMN* (PF00724/IPR001155) domain were identified as *OPRs*. Besides, *OPR* genes from *Gossypium arboreum*, *Gossypium raimondii*, *Helianthus annuus*, *Oryza sativa*, *Solanum lycopersicum*, *Triticum aestivum*, *Zea mays*, and *Arabidopsis thaliana* were identified by the same method.

### 2.2. Phylogenetic Analysis

Amino acid sequences of *OPRs* identified in *G. hirsutum*, *G. barbadense*, and *OPR* genes from other plant species were subjected to multiple sequence alignment using MUSCLE [34]. After conducting a model test, a maximum likelihood (ML) phylogenic tree was constructed with the best substitution model using MEGA X software [35].

### 2.3. Gene Structure Analysis and Conserved Motif Identification

Exon-intron structure information of *OPR* genes and coordinates of characteristic domain were extracted from reference gene annotation data and InterProscan annotation results. Gene Structure Display Server (GSDS) v2.0 [36] was used to display gene structure and indicate characteristic domain coding regions. Conserved motifs were identified using the Multiple Expectation Maximization for Motif Elicitation (MEME) program v.5.0.5 [37] with the default parameters.

### 2.4. Genomic Distribution, Collinearity and Duplication Analysis of Cotton OPR Genes

The genomic coordinates of cotton *OPR* genes were extracted from genome annotation data, and RIdeogram [38] was used to display genomic distribution of cotton *OPR* genes. MCScanX [39] was used to identify genomic collinear blocks and tandem duplications with the default parameters, and collinearity relationship was visualized by Circos v.0.69 [40].

### 2.5. Promoter and Regulatory Analysis of Cotton OPR Genes

The upstream 1.5 kb sequences of gene coding region of *OPR* genes were extracted as promoter regions. Promoter region sequences were submitted to the PlantCARE database (http://bioinformatics.psb.ugent.be/webtools/plantcare/html/) to conduct the prediction of cis-acting elements. The cis-acting elements distribution upon promoters of OPR genes were displayed by GSDS v2.0 (http://gsds.cbi.pku.edu.cn./) [36].

### 2.6. Plant Materials

The resistant *G. hirsutum* cv. Zhongzhimian No. 2 to *V. dahliae* and the susceptible *G. hirsutum* cv. Jimian No.11 were the plant materials. Plants were grown in a growth chamber with 16 h day/8 h day/night cycle at 25 °C.

### 2.7. Treatment with PEG6000, NaCl and V. dahliae

Seedlings of *G. hirsutum* cv. Zhongzhimian No. 2 were used for gene expression in response to different stresses. For the abiotic stresses (drought and salt), three-leaf-stage seedlings were cultivated in a Hoagland liquid medium including 20% PEG6000 and 200 mM NaCl, respectively. The leaves of cotton were collected at four time points (0, 3, 6, and 12 h) for RNA extraction. *V. dahliae* infection was performed as described previously [41]. The seedling leaves, stems, and roots were harvested at eight time points (0, 1, 3, 6, 9, 12, 24, and 48 h) after Vd080 infection for RNA extraction, respectively. At least 50 plants were treated by each experiment. Each time point samples contained three plants, the samples were quickly frozen in liquid nitrogen and stored at −80 °C. The experiment was replicated twice.

### 2.8. VIGS

The technology of virus-induced gene silencing was performed as reported previously [42]. Fragments of *GhOPR9* and *GhPDS* for evaluating VIGS marker gene were amplified from *G. hirsutum* cv. Zhongzhimian No.2 cDNA by PCR, and then integrated into the tobacco rattle virus (TRV) vector pYL156 at the *Xba*I-*Sac*I sites using the In-Fusion HD Cloning Kit (Clontech, Mountainv View, CA, USA) according to the manufacturer’s protocol. Then, the plasmids of pYL-*GhOPR9*, pYL-*GhPDS*, the pYL156 empty vector, and the auxiliary vector pLY192 were transformed into *Agrobacterium tumefaciens* strains GV3101. A positive *A. tumefaciens* single colony of each vector was cultured with 50 µg/mL kanamycin and 50 µg/mL rifampicin in LB medium at 28 °C for 12–16 h. The Agrobacterium cells were collected by centrifugation. Following, they were resuspended to a ~1.0 value of OD_600_ via using MMA solution (10 mM N-morpholino ethanesulfonic acid, 10 mM MgCl_2_ and 200 mM acetosyringone). The Agrobacterium cells of pYL-*GhOPR9* (TRV:*GhOPR9*), pYL-*GhPDS* (TRV:*PDS*), and pYL156 empty vector (TRV:*00*) were mixed with the Agrobacterium strains of pLY192 in a 1:1 ratio, respectively; they were incubated at 28 °C for 3 h in the dark. Finally, the mixture cells of TRV:*PDS*, TRV:*00*, and TRV:*GhOPR9* were infected into the cotyledons of which were fully expanded seedlings via needleless syringe. At least 50 plants were injected by each construct. Three plants were collected from each treatment at each time point. The experiment was replicated twice. When the newly true leaves of TRV:*PDS* lines presented a photobleaching phenotype, the silencing efficiency of the target gene *GhOPR9* in TRV:*GhOPR9* lines was examined using the newly true leaves via RT-qPCR, the TRV:*00* lines were the control. The successfully silenced plants were used to inoculate with *V. dahliae* strain Vd080. The TRV:*00* lines with Vd080 inoculation were the control.

### 2.9. Pathogen Infection and Disease Assay

The highly aggressive defoliating Vd080 was cultured as described previously [43]. The fungus was grown on potato dextrose agar medium at 25 °C for seven days in a dark place. Then, the highly activated hyphae were collected and cultured in potato dextrose broth medium at 25 °C for five days in dark place. The conidia of *V. dahliae* were resuspended in distilled water. The final concentration of 10^7^ spore mL^−1^ was used for infection. Fungal inoculation was conducted as described previously [41]. The wild type (WT) cotton seedlings were infected by Vd080 to make sure that the species can be infected with Vd080 and present the typical symptom of *V. dahliae*, and the WT seedlings infected with water were the control. The plants roots were dipped into 1 × 10^7^ spore mL^−1^ conidial suspension for 10 min. The successful silencing plants of TRV:*GhOPR9* were infected with conidial suspension. After inoculation, cotton leaves and roots were collected at four time points (00, 01, 12, and 24 h) and washed with water for RNA extraction to detect the expression levels of resistance genes in JA pathway. A disease index (DI) was calculated as described previously [44]. Seedling stems were cut from each line at the same position to investigate the vascular wilt symptom via a microscope. The fungal DNA abundance assay was performed as described previously [27]. Seedling stems were collected at three weeks post-inoculation. Total DNA was extracted for qPCR to detect the fungal biomass. The *ITS1-F*/*ST-VE1-R* primer were used to detect the fungal DNA. The TRV:*00* lines with Vd080 inoculation were the control. The cotton gene *GhUB7* was used as control for qPCR analysis. Each experiment was replicated twice. The primers used in this study are listed in Appendix A.

### 2.10. Callose Deposition

Callose depositions were visualized by aniline blue staining as described previously [45]. Leaf sample were first destained in 3:1 ethanol/acetic acid for 3 h and were then soaked into 70% and 50% ethanol for 2 h, respectively. Following, leaves were transferred into the water for 12 h. And then, the leaves were destained in 10% (w/v) NaOH for 2 h. Finally, they were stained with 0.01% (w/v) aniline blue in 150 mM K_2_POH_4_ (pH9.5). Stained leaves were imaged on fluorescence microscopy. At least three leaves of TRV:*00* and TRV:*GhOPR9* lines at three weeks post-inoculation were observed, and each experiments were replicated twice.

### 2.11. RNA/DNA Extracted and Real-time Quantitative PCR/Quantitative PCR Analysis

Total RNA was extracted from the collected samples using the RNAprep Pure Plant Plus Kit (Polysaccharides & Polyphenolics-rich) (TransGen Biotech, Beijing, China) according to the manufacturer’s instructions. The first cDNA strand was synthesized by using the All-in-One First-Strand cDNA Synthesis Super Mix for qPCR Kit (One-Step gDNA Removal) (TransGen, Beijing, China) according to the manufacturer’s instructions. The RT-qPCR was performed using the Roche Light Cycler 480 System (Roche, Mannheim, Germany). A 20-µL reaction system was used, the components of reaction were displayed as follow: 2 µL (200 ng) of cDNA, 0.4 µL of forward primer (10.0 µmol/L), 0.4 µL of reverse primer (10.0 µmol/L), 10 µL of 2 × TransStart Top/Tip Green qPCR Super Mix, and 7.2 µL of nuclease-free water. The reaction procedure was completed as the following program: 94 °C for 30 s; 45 cycles of 94 °C for 5 s, 60 °C for 15 s, 72 °C for 10 s; and 4 °C for ending. Total DNA was extracted from the collected samples using the Fungal DNA Kit (Omega Bio-tek, Norcross Georgia, US according to the manufacturer’s instructions. The qPCR of DNA samples was performed as above describe. Primer sequences used in the present study are presented in Appendix A. The expression levels of *OPR* genes were analyzed by the 2^−ΔΔCT^ method.

## 3. Results

### 3.1. Genome-Wide Identification of OPR Genes Family in Cotton

We identified 10, 9, 5, and 3 *OPRs* by detecting the *Oxidored_FMN* (ID:PF00724) conserved domain via Pfam in *G. hirsutum* (ZJU), *G. barbadense* (ZJU), *G. arboreum*, and *G. raimondii*, respectively. The gene ID, genomic location, DNA length, coding sequence length, protein length, molecular weight (MW), and isoelectric point (IP) are presented in Appendix A. The lengths of OPR proteins ranged from 156 to 418 amino acids, the MW ranged from 17.476 kDa to 46.634 kDa, and the IP ranged from 4.914 to 8.749.

### 3.2. Phylogenetic Analysis of OPR Genes

To further investigate the phylogenetic relationships of the *OPRs* in different species, we used the 10 *OPRs* from *G. hirsutum*, nine *OPRs* from *G. barbadense*, five *OPRs* from *G. arboreum*, three *OPRs* from *G. raimondii*, 22 *OPRs* from *Helianthus annuus*, four *OPRs* from *S. lycopersicum*, three *OPRs* from *A. thaliana*, 48 *OPRs* from *T. aestivum*, 12 *OPRs* from *O. sativa*, and eight *OPRs* from *Z. mays* for constructing a phylogenetic tree by MEGA X software using the ML method. The phylogenetic tree divided 124 *OPRs* into five different groups (Figure 1). The different species *OPRs* named were displayed in Appendix A.

The result showed that Group I was the biggest, which had 54 OPR proteins. Group V was the smallest, which gathered only 11 *OPRs*. Group II, Group III, and Group IV contained 21, 16, and 22 OPR proteins, respectively. Monocots species OPR members were clustered in all five Groups, such as Ta*OPRs*, which was consistent with previous studies [20]. Dicots species *OPRs* were distributed in Group I and Group II. Interestingly, most of dicots *OPRs* were tightly related to AtOPR1 and AtOPR2 and were gathered in Group I, contain 20 *HaOPRs*, 17 *OPRs* from four different species cotton and three *SlOPRs*. Group IV contain 10 OPR members of cotton, two *HaOPRs*, and one SlOPR and tightly linked to AtOPR3. Compared to other species, cotton *OPRs* have a closer relationship with the *AtOPRs*. GaOPR4, GbOPR4, GbOPR7, GbOPR8, GhOPR4, GhOPR7, GhOPR8, and GhOPR9 were clustered in the same clad with AtOPR1 and AtOPR2. GaOPR5, GbOPR3, GbOPR6, GbOPR9, GhOPR3, GhOPR6, GhOPR10, and GrOPR3II showed a closely relationship with the AtOPR3. A previous study showed that *OPR3* plays a key role in JA biosynthesis in *A. thaliana* [14]. The functions of *OPRs* that are tightly related to *AtOPR1* and *AtOPR2* in other species are still not well understood. Their contributions could not be ignored. Therefore, there were difference in the functions of the *OPR* genes between monocotyledons and dicotyledons. Furthermore, the protein of *OPRs* had the conserved domains, but their functions might be diverse.

### 3.3. Gene Structural, Conserved Motif Analysis of OPRs in Gossypium

To better understand the phylogenetic relationships and gene structures of the *OPR* family in cotton, we used the 10 *OPRs* and nine *OPRs* from *G. hirsutum* and *G. barbadense*, respectively, and three *OPRs* from *A. thaliana* for constructing a phylogenetic tree by MEGA X software (Figure 2a). The result showed that the relationships of the *OPRs* were consist with the phylogenetic tree in Figure 1. The OPR proteins from cotton were divided into three subgroups. GbOPR4, GbOPR7, GbOPR8, GhOPR4, GhOPR7, GhOPR8, and GhOPR9 were gathered in the same clad with AtOPR1 and AtOPR2. GbOPR3, GbOPR6, GbOPR9, GhOPR3, GhOPR6, and GhOPR10 were clustered in a same clad with the AtOPR3. Meanwhile, GbOPR1, GbOPR2, GbOPR5, GhOPR1, GhOPR2, and GhOPR5 were grouped into another clade.

We analyzed the exon-intron structures of *OPR* genes. 12 *OPRs* in cotton contained five exons and four introns. Three *OPR* genes contained four exons and three introns: *GhOPR2*, *GbOPR9* and *GhOPR10*. Two *OPR* genes possessed three exons and two introns: *GbOPR5* and *GhOPR5*. *GbOPR1* and *GhOPR1* have the most exons and introns—six and five, respectively. As presented in Figure 2b, the intron or exon numbers and lengths in different subgroups were different. Loss or gain of the exon/intron took place during the evolution of *OPR* genes family in cotton, especially in sub. II. Our results suggested that *OPR* genes maintained a relatively constant exon-intron composition during evolution of the *Gossypium* genome.

To estimate the conserved motif of OPR proteins in *Gossypium*, MEME analysis was performed to predict distinct motifs. Fifteen putative motifs named motifs 1–15, were finally identified. These motifs contained varied from 6 to 50 amino acids, and the details of the 15 conserved motifs are displayed in Appendix A. As shown in Figure 2c, most of the orthologous proteins shared similar motif members in the same subgroup. In subgroup I, there have motif 1–9 and motif 11 in each OPR. Compared to other proteases, GhOPR9 lacked one motif 6, instead of a motif 12. In sub. II, the motifs displayed different patterns, GbOPR1 and GhOPR1 had the same motifs 1–8, 12; GbOPR2 and GhOPR2 lacked motifs 5 and 7, but gain a motif 10; GbOPR5 and GhOPR5 lacked motif 4 and motif 7, also have motif 10. In sub. II, *OPRs* shared motif 1–9 and 13–15 in relatively conserved pattern. In addition, each class have the specific motifs, such as motif 11 in sub. I, motif 12 in sub. II, motif 13–15 in sub. III. Among of the motifs, motif 1, motif 2, motif 3, motif 4, motif 5, and motif 6 were corresponded to the beta/alpha barrel which might be related to the protein secondary structures [20]. In conclusion, the results provided an additional evidence to support the classification results.

### 3.4. Chromosomal Location and Gene Synteny Analysis of OPR Genes in Gossypium

The distribution of *OPR* genes was investigated by positioning their approximate positions on cotton chromosome. As shown in Appendix A, chromosomes GH_A01, GH_A05, GH_D01, GH_D05, and GH_D06 contained all of the *GhOPR* genes; and chromosomes GB_A01, GB_A05, GB_D01, GB_D05, and GB_D06 contained all of the *GbOPR* genes. In *G. hirsutum*, chromosome GH_D05 have the most *GhOPR* genes (four, 40%). Moreover, two OPR genes were distributed in chromosomes GH_A01 and GH_A05 and only one gene in chromosomes GH_D01 and GH_D06. Furthermore, the genes in chromosome GH_A05 and GH_D05 were primarily anchored on chromosome head. In *G. barbadense*, the distribution of *OPR* genes was almost the same as the *G. hirsutum*. However, compared to GH_D05, GB_D05 lacked one *OPR* gene.

To further understand the relationships of *OPR* genes between *G. hirsutum* and *G. barbadense*, syntenic analysis were performed (Figure 3). Among the candidate *OPR* genes, eight *GhOPRs* were the orthologous genes of the eight *GbOPRs*. Some *OPRs* of *G. hirsutum* had not only one orthologous gene in *G. barbadense*, such as *GhOPR2* and *GhOPR5*, which had two genes (*GbOPR2* and *GbOPR5*); *GhOPR3*, *GhOPR6*, and *GhOPR10* had three genes (*GbOPR3*, *GbOPR6*, and *GbOPR9*); and *GhOPR4* and *GhOPR7* had two genes (*GbOPR4* and *GbOPR7*). In addition, there were three paralogous gene pairs (*GhOPR2/5*, *GhOPR3/6/10* and *GhOPR4/7*) in *G. hirsutum* genome, and three paralogous gene pairs (*GbOPR2/5*, *GbOPR3/6/9* and *GbOPR4/7*) in *G. barbadense* genome (Appendix A). Furthermore, seven *GhOPRs* and seven *GbOPRs* were classified into WGD or segmental duplications (Appendix A), and one *OPR* member was classified into dispersed duplication in *G. hirsutum* and *G. barbadense*, respectively. It was noteworthy that two *OPRs* were tandem duplication genes in *G. hirsutum*. WGD or segmental duplication might play a crucial role in the expansion of the *OPR* gene family.

### 3.5. Prediction of Cis-Acting Elements in the Promoters of OPRs in G. hirsutum

To further predict the possible biological functions of *GhOPRs*, the 1.5 kb upstream promoter regions of all *GhOPRs* were obtained and analyzed the cis-acting regulatory elements via the online database PlantCARE. Thirty kinds of elements were discovered in the promoter regions of *OPR* genes in *G. hirsutum* (Appendix A). The cis-acting elements which were the binding regions of transcription factors played a crucial role in regulating gene expression. As shown in Appendix A, a large number of cis-acting elements were predicted to be related to transcription, various hormones and stresses response, cell cycle, and development. Four kinds of elements that related to the core cis-acting element, light response, and wounding response were predicted in all *GhOPR* genes promoter regions. The number of the core cis-acting elements was 682 in *G. hirsutum*, which was the most cis-acting element. In particular, a specific element might be related to binding site, MYBHv1 binding site was found in *GhOPR10* promoter region, a cis-acting element predicted to be related to circadian was just only discovered in promoter region of *GhOPR6*. The cis-acting elements might be related to hormone signaling pathways and response to various stresses were predicted in *GhOPRs* promoter regions, for instance ABA, auxin (IAA), ET, gibberellins (GA), JA, methyl jasmonate (MeJA), SA, cold, drought, wounding, pathogen, etc. Interestingly, a JA response cis-acting element was just found in *GhOPR9* promoter region (Table 1), indicating that *GhOPR9* might participate in JA-mediated signaling pathways. These results indicated that *OPR* genes in *G. hirsutum* might perform different biological functions.

### 3.6. Expression Patterns of GhOPRs in Response to Abiotic Stresses

To further understand expression levels of the *GhOPRs* under drought and salt stresses, RT-qPCR was performed using the leaves of upland cotton Zhongzhimian No. 2 treated with PEG6000 and NaCl (Figure 4). The results showed that all of *GhOPRs* were repressed by PEG6000 and induced by NaCl, respectively. Under the PEG stress, the transcription levels of *GhOPRs* exhibited a decreasing trend over 00–12 h ranges in leaves, except for *GhOPR7*. In subgroup I and subgroup III, *GhOPR* genes expressions were down-regulated at 06 h, and then, up-regulated at 12 h, but were still lower than 00 h. Under NaCl treatment, all of *GhOPRs* expression levels were significantly increased. Eight of these *GhOPRs* reached peak transcription levels at 06 h, and then, decreased at 12 h, but still were higher than control. In addition, *GhOPR3*, *GhOPR6*, and *GhOPR10* got the peak expression levels at 12 h. Furthermore, paralogous genes had the same transcription patterns in response to PEG or NaCl treatment, such as *GhOPR2/5* and *GhOPR4/7*. These results suggested that the *GhOPRs* might play an important role in response to abiotic stresses in cotton.

### 3.7. Expression Patterns of OPR Genes in G. hirsutum Under V. dahliae Inoculation

To further ascertain whether *GhOPRs* RNA levels were related to *V. dahliae* infection, we performed RT-qPCR to analyse the expression profiles of *OPR* genes using leaf, stem and root of Zhongzhimian No. 2 under *V. dahliae* inoculation during 48 h. The relative expression of *GhOPRs* showed various expression profiles (Figure 5).

In *G. hirsutum*, nine *OPRs* were decreased at 1 h, followed by an increase in leaves, except *GhOPR1* (Figure 5a). In subgroup I and subgroup II, most expression levels of *GhOPRs* got the peak at the 12 h and then were downregulated. In subgroup III, *GhOPR3*, *GhOPR6*, and *GhOPR10* were highly expressed. *GhOPR3* and *GhOPR6* were up-regulated from 1 h to 6 h and down-regulated from 9 h to 24 h in leaves, and the highest expression levels were reached at 48 h. All *OPRs* were significant induced in root (Figure 5b). In subgroup I and subgroup II, our results showed that the expression levels of *GhOPR4*, *GhOPR7*, *GhOPR8*, *GhOPR9*, *GhOPR1*, *GhOPR2*, and *GhOPR5* were significantly increased in root. *GhOPR5* got the highest expression level at 48 h, and the other six *GhOPRs* gained their highest expression levels at 24 h. The expression levels of *GhOPR9*, *GhOPR1*, *GhOPR2* and *GhOPR5* were almost sustainably increased during the process under *V. dahliae* inoculation in root in 24 h. In particular, *GhOPR9* increased more than 100 times at 24 h. But in sub. III, the genes were downregulated from 01 to 24 h after *V. dahliae* inoculation. All of *OPRs* received their highest expression levels at 12 h in stem (Figure 5c). The *GhOPR9* was evidently upregulated by four times at 24 h. The *GhOPRs* were significantly higher induced by *V. dahliae* in root than in leaf and stem. The *GhOPRs* expression patterns were similar in same subgroup in each tissue. These results suggested that the *GhOPRs* might be play an important role in response to *V. dahliae* infection in cotton.

### 3.8. Silencing GhOPR9 Attenuates the Resistance of Cotton to V. dahliae

Expression profiles of *GhOPRs* showed that *GhOPR9* was significantly induced in cotton under *V. dahliae* infection. To investigate the role of *GhOPR9* in Verticillium wilt resistance, TRV-based VIGS was performed using resistant *G. hirsutum* cv. Zhongzhimian No.2. After cotton growing 10 days, the code sequences of *GhOPR9* which about 250 bp were integrated into the vector pTRV2 (TRV:*GhOPR9*) for cotton seedlings infection. The fragment of *GhPDS* (TRV:*GhPDS*) or the pTRV2 empty vector (TRV:*00*) was performed as a VIGS indicator or a control to infection the plants cotyledon. When the newly true leaves that were infected with TRV:*GhPDS* showed a photobleaching phenotype (Figure 6a), the gene-silenced efficiency of TRV:*00* and TRV:*GhOPR9* plants was performed by RT-qPCR. RT-qPCR results showed that the gene silencing of *GhOPR9* was successful (Figure 6b). Then, the gene-silenced plants were infected with Vd080 at the three leaf-stage. The TRV:*00* plants were infected with Vd080 as control. The typical disease symptoms such as wilting, chlorosis, necrosis and darken vascular bundles was present on the WT cotton plants under Vd080 infection (Figure 6c,d). Compared to the TRV:*00* plants, *GhOPR9*-silenced plants displayed more severely symptoms than the control (Figure 6c,e). The results displayed that *GhOPR9*-silenced plants significantly impaired the resistance to Vd080. The disease index analysis and fungal biomass detection of *GhOPR9*-silenced lines were much higher than the TRV:*00* lines (Figure 6f,g). Moreover, *GhOPR9*-silenced lines displayed weakly callose deposition after infection with Vd080, compared to the TRV:*00* plants (Figure 6h). These results suggested that silencing of *GhOPR9* can increase the susceptibility of cotton to *V. dahliae* infection.

### 3.9. GhOPR9 Modulates Expression of JA-Regulated Defence Genes under V. dahliae Inoculation

To test whether knock-down of the *GhOPR9* gene impacted the expression of JA pathway-regulated genes under the *V. dahliae* infection or not, RT-qPCR were performed in the background of cotton Zhongzhimian No.2 in which *GhOPR9* was silenced after *V. dahliae* inoculation, TRV:*00* lines under the *V. dahliae* inoculation were the control. *LOX*, *AOS* and *AOC* are the upstream genes of *OPR2* in JAs biosynthesis pathway [46]. The *GhLOX2*, *GhLOX3*, *GhLOX4*, *GhLOX6*, *GhAOS*, *GhAOC*, and the *OPR3* of *G. hirsutum* were selected for the candidate genes. As Figure 7 shows, *GhAOS* and *GhAOC* transcriptional levels were increased the peak levels at 01 h but decreased at 12 h and 24 h in roots (Figure 7b). Compared to the TRV:*00* lines, the expression levels of *GhAOS* and *GhAOC* were lower in the TRV:*GhOPR9* plants at 1 h, 12 h, and 24 h. The expression level of *GhLOX2* in TRV:*GhOPR9* plants leaves was higher than TRV:*00* lines at 01 h, and was lower at 12 h and 24 h (Figure 7a). On the contrary, the *GhLOX2* expression level in TRV:*GhOPR9* plants roots was lower than TRV:*00* lines at 1 h, and was higher at 12 h and 24 h (Figure 7b). In contrast, the expression levels of *GhLOX3* and*GhLOX6* had increasing trends in leaves (Figure 7a), but the *GhLOX3* expression level was increased at 01 h and decreased from 12 h to 24 h in roots; the *GhLOX6* expression level had a decreasing trend in roots (Figure 7b). In addition, these genes expression levels in TRV:*GhOPR9* plants roots were lower than that in TRV:*00* lines after *V. dahliae* inoculation, except the *GhLOX2* expression pattern at 01 h (Figure 7b).

## 4. Discussion

The *OPR* gene family was extensively presented in plants [21]. They had been well investigated in many plants. They were identified to be related to different physiological and biological functions. In rice, 13 *OPR* genes were uncovered, the *OsOPR* genes expression profiles were different in different tissues. The expression profiles were also different under different stresses. The *OsOPRs* might play multiple physiological and biological roles in rice [19]. In wheat, 48 *OPR* genes were identified. The *TaOPRs* showed diverse expression levels in response to various stresses, such as biotic (aphid), abiotic stress (wounding, salt, heat, drought) and exogenous hormone treatments (MeJA, ABA and JA) [20]. Among of the *TaOPRs*, *TaOPR1* was involved in ABA signaling pathway and could enhance the reactive oxygen species scavenging to confer salinity tolerance [22]. *TaOPR2* was related to the biosynthesis of JA [23]. In *Z. mays*, the maize genome encoded 8 *OPR* genes. *ZmOPR1*, *ZmOPR2*, *ZmOPR3*, *ZmOPR4*, and *ZmOPR2* were in Group I, *ZmOPR5*, *ZmOPR7* and *ZmOPR8* were in Group II [16]. In cotton, previous study showed that the OPR3 of *G. hirsutum* was phosphorylated by GhCPK33 to suppress JA accumulation and JA signaling when host was infected by *V. dahliae* [27], and the other *OPR* genes were not studied explicitly. Though a genome-wide analysis of *OPRs* in cotton, we identified 10 *OPR* genes in *G. hirsutum*. According to our phylogenetic analysis, *GhOPR* genes were divided into two major groups in consistent with *A. thaliana* [14], and Group I was further classified into two subgroups in *G. hirsutum*. Compared to dicots, *OPR* genes were divided into five groups in monocots, such as wheat [20]. The previous study showed that lineage-specific expansion events occurred between higher and lower land plants [21]. Similarly, lineage-specific expansion events also occurred in higher land monocots (wheat), and extra three groups were generated after the divergence from dicots (cotton). Combined with the gene structure analysis, most of *OPR* genes in a group showed a similar exon/intron structure, indicating that the evolution might affect both gene function and gene structure [47,48].

In fact, not only the structural diversity of gene family members was a mechanism for the evolution of multiple gene families, but also intron loss and gain could play an important role in generating structural diversity and complexity [49]. In this study, only *GhOPR5* had two introns, and *GhOPR10* had three introns, most of other *OPR* genes contained 4–6 introns. However, 33 of 48 *TaOPRs* had less than three introns [20]. *ZmOPR1*, *ZmOPR2*, and *ZmOPR3* had only one intron [16]. These results revealed that the intron loss events occurred in different plant lineages from cotton to wheat or maize. Furthermore, the intron loss and gain occurred from ancestral *OPR* genes to present individual *OPR* genes in individual plant lineage. As the same with the results of the gene structure analysis, most of *OPRs* in a subgroup shared conserved protein motifs. Almost all of the *OPRs* had the motif 1, motif 2, motif 3, motif 4, motif 5, and motif 6. They were corresponded to the beta/alpha barrel and might be related to the OPR conserved protein domain [20]. However, some OPR contained specific motif; for instance, in subgroup I, GhOPR1, GhOPR2, GhOPR5, and GhOPR9 had motif 12. Interestingly, these *OPRs* had the same expression patterns in leaves and roots under Vd080 infection. Among of 4 *GhOPRs*, *GhOPR9* expression level was significantly increased. These results suggested that diversity of motif member and gene structure probably contributed to gene expression under stress. In order to respond to various stresses, the *OPR* genes probably formed various gene structures or protein motifs during the genome evolution in cotton under the selection pressures. Tandem duplication, segmental duplication and genome duplication contributed to expansion of gene families and genome evolution [50,51]. As the results showed, the collinear relationships of *OPR* genes were strong in cotton. We found 7 WGD or segmental duplication *OPR* genes in *G. hirsutum* and *G. barbadense*, respectively. In comparison with the WGD/segmental duplication, we just found 2 tandem duplication *OPR* genes in *G. hirsutum*, *GhOPR8* and *GhOPR9*. But in *G. barbadense*, there was no tandem duplication *OPR* gene. In this study, WGD/segmental duplication possible made more contribution to expansion of OPR gene family than the tandem duplication, and tandem duplication genes contributed to the new biological functions in cotton genome evolution [52].

According to the prediction of cis-acting regulatory elements, *GhOPR* might play a role in regulating various biological processes in cotton. The *GhOPRs* might have association with phytohormones stimulation, such as JA, MeJA, ABA, GA, SA, IAA, and ET. It could be explained that *OPR* genes were involved in various signal pathways to participate in plant growth, development, and defensive responses. In wheat, *TaOPR1* and *TaOPR2* were involved in ABA and JA signal pathways, respectively, and could be induced by various stresses [22,23]. The *SlOPR3-RNAi* plants failed to accumulate JA after wounding and reduced trichome formation and affected monoterpene and sesquiterpene production; thus, the defense of tomato (*S. lycopersicum*) to the specialist herbivore *Manduca sexta* was reduced [53]. These could be suggested that *OPRs* might joined in stress resistance via hormone signaling pathways. In this study, the *GhOPRs* were down-regulated in response to PEG stress, and up-regulated in response to NaCl stress. But in wheat, the *TaOPRs* were down-regulated under salt stress [20]. The *OPRs* genes exhibited opposite expression profiles under salt treatment, the results indicated that the functions of *GhOPRs* might be different between dicots and monocots under abiotic stresses. Under the biotic stress, all of *GhOPRs* were response to the Vd080 infection in leaf, root and stem. And the *GhOPR9* expression was significantly upregulated in root. These results indicated that *OPRs* regulated various responses to abiotic and biotic stresses, the mechanism of *OPRs* functions needed to be further investigated.

Verticillium wilt is the most destructive disease of cotton. About 200 dicotyledonous plant species are susceptible to this notorious pathogen. To date, there is no efficient chemical pesticide available for cotton Verticillium wilt, such as *G. hirsutum* as the cultivated cotton species, there are few germplasms that are immune or highly resistant to *V. dahliae* [54,55]. The battle between plants and pathogens promotes the evolution of various defense pathways in the host and the attack strategies of pathogens. In particular, JAs, SAs, and ET, as the essential defensive roles of primary defense hormones, have been well understood [56]. The elevation of JA accumulation is usually occurred within a few minutes or a few hours after wounding or pathogen infection [57]. In this study, the *GhOPRs* expression levels were decreased at 1 h after Vd080 infection in cotton leaves, except *GhOPR1*. Previous study showed that the antagonistic crosstalk between the JA and SA pathways were effective against pathogens [58]. In this study, *GhOPRs* expression levels displayed fluctuation in leaves from 0 h to 48 h after Vd080 inoculation. These results might be related to the crosstalk of JA and SA pathways. Previous study showed that JA signaling can promote susceptibility to hemibiotrophic pathogens [59]. And in cotton, JA is the key hormone regulating response to *V. dahliae* [60]. VIGS is a quick and powerful technique to assess the function of genes by transient post-transcriptional gene silencing [61]. Agrobacterium-mediated TRV-VIGS could be used in cotton leaves and fiber [62,63]. Silencing *GhNDR1* and *GhMKK2* via TRV-VIGS technique compromised the cotton resistance to Verticillium wilt [64]. Silencing *GhCPK33* induced JA accumulation and enhanced resistance to *V. dahliae* [27]. *GbWRKY1*, *GhCYP82D*, *GhLac1*, and *GhJAZ2* could modulate resistance to *V. dahliae* in cotton by regulating JA biosynthesis [65,66,67,68]. JA response-associated genes were upregulated in *GhWRKY70*-silenced cotton plants and could increase the resistance of cotton to *V. dahliae* [69]. In the present study, *GhOPR9* silenced plants showed susceptibility to Vd080 infection, moreover, the expression levels of *GhLOX3*, *GhLOX4*, *GhLOX6*, *GhAOS*, *GhAOC*, and *OPR3* of *G. hirsutum*, which were the upstream genes of *OPR2* in silenced plants roots, were lower than the expression levels of these genes in TRV:*00* plants root after Vd080 infection. These suggested that *GhOPR9* might positively regulate the resistance of cotton to *V. dahliae* via mediated the JAs pathway genes. Furthermore, in susceptible *G. hirsutum* cv. Jimian No.11 (Appendix A), *GhOPR9*-silenced lines displayed more susceptibility to Vd080 than the TRV:*00* plants. Considering the tandem duplication of *GhOPR9*, *GhOPR8* was also silenced by VIGS in Zhongzhimian No. 2 (Appendix A). These tandem duplication genes maybe had the functional redundancy in regulating the JAs related genes to modulate the resistance of cotton to *V. dahliae*. However, the precise mechanism of the *GhOPR9* function remains to be further studied.

## 5. Conclusions

In this study, a total of 19 *OPR* genes were identified in *G. hirsutum* and *G. barbadense*, which were classified into three groups. WGD or segmental duplication might play the principal role in the expansion of the *OPR* gene family in cotton. The cotton *OPRs* might be related to crucial processes—for instance, plant growth and development, phytohormone signal pathway, and defensive responses to various abiotic and biotic stresses. The expression patterns of *OPR* genes had PEG, NaCl, and *V. dahliae* stress-responsive diversity. Furthermore, *GhOPR9* positively regulated the resistance of cotton to *V. dahliae* via mediating the JAs pathway genes. This study can help us to have a better understanding of the *OPR* genes in cotton and can also help us to screen candidate genes with high resistance to *V. dahliae*.

## Figures and Tables

**Figure 1 genes-11-01134-f001:**
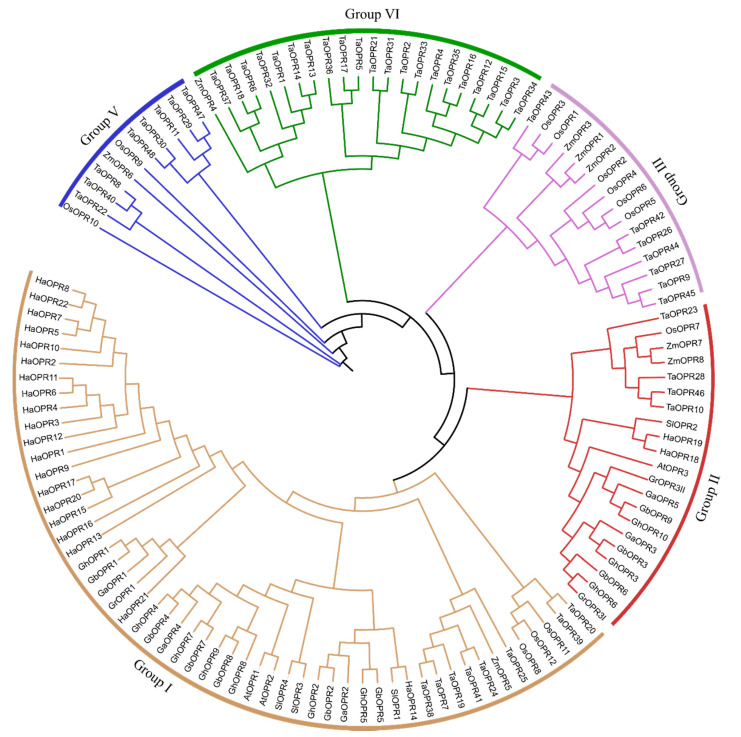
The maximum likelihood (ML) phylogenetic tree of the 12-oxo-phytodienoic acid reductase (*OPR*) family. The tree was drawn with the full-length amino acid sequences of *OPR* genes from *Arabidopsis thaliana* (Linn.) Heynh. (At), *Gossypium arboreum* L. (Ga), *G. barbadense* L. (Gb), *G. hirsutum* L. (Gh), *G. raimondii* Ulbr. (Gr), *Helianthus annuus* L. (Ha), *Oryza sativa* L. (Os), *Solanum lycopersicum* L. (Sl), *Triticum aestivum* L. (Ta), and *Zea mays* L. (Zm), using MEGA X, with 1000 replicates. They were classified into five groups. Group I, Group II, Group III, Group VI, and Group V are represented by orange, red, purple, green, and blue, respectively.

**Figure 2 genes-11-01134-f002:**
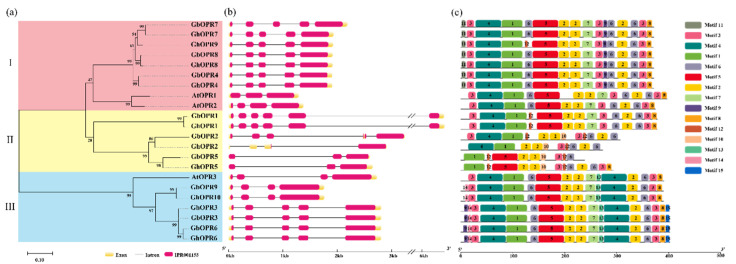
Structural and motif analysis of two allotetraploid cotton *OPR* genes. (**a**) Phylogenetic tree of *OPR* genes in *G. barbadense* and *G. hirsutum*. The ML phylogenetic tree was constructed by using MEGA X with 1000 replicates. (**b**) Exon-intron structures of *GhOPR* and *GbOPR* genes. Orange boxes mean exons, and black lines mean introns, and red color represent conserved domain. (**c**) Conserved motifs of GhOPR and GbOPR proteins. Fifteen conserved motifs are represented by different color boxes.

**Figure 3 genes-11-01134-f003:**
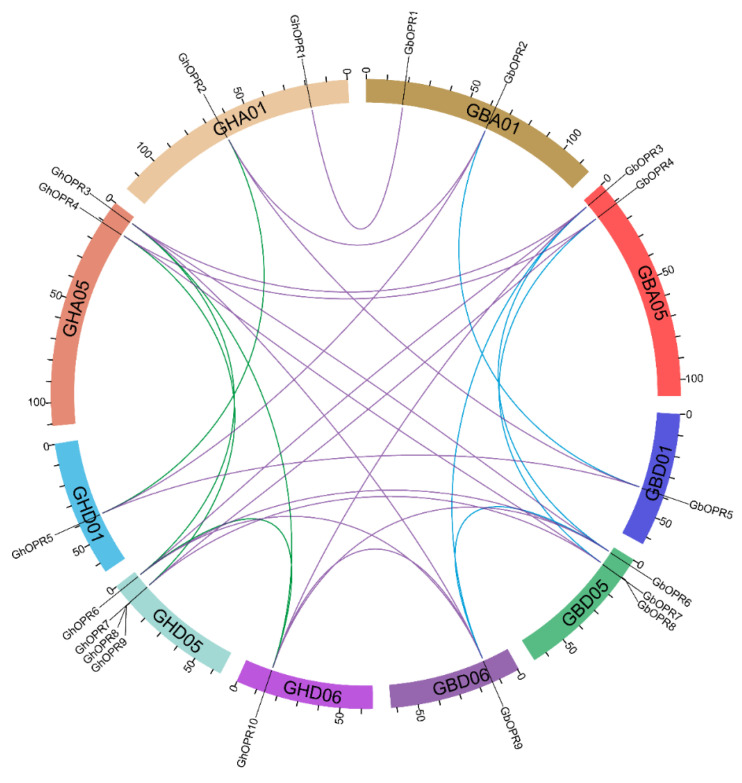
The synteny relationship of *OPR* genes between *G. hirsutum* and *G. barbadense*. Five upland cotton and five sea-island cotton chromosomes are displayed with different random colors. Lines in purple represent orthologous gene pairs, green represent upland cotton paralogous gene pairs, and blue represent sea-island paralogous gene pairs.

**Figure 4 genes-11-01134-f004:**
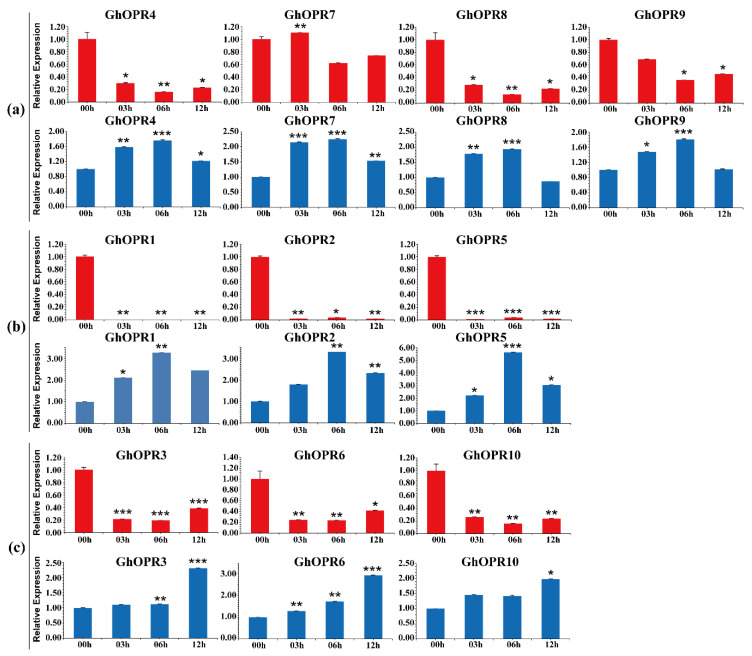
Analysis expression patterns of *GhOPRs* under abiotic stresses by Real-time quantitative PCR. (**a**) The expression levels of *GhOPRs* in subgroup I. (**b**) The expression levels of *GhOPRs* in subgroup II. (**c**) The expression levels of *GhOPRs* in subgroup III. The red columns represent the expression levels of *GhOPRs* under PEG6000 treatment; and blue columns represent the expression levels of *GhOPRs* under NaCl treatment. Values represent means ± standard deviation of three replicates. Asterisks reveal the gene significantly higher or lower in 3 h, 6 h, and 12 h than in 0 h by *t*-test (* *p* < 0.05, ** *p* < 0.01, *** *p* < 0.001).

**Figure 5 genes-11-01134-f005:**
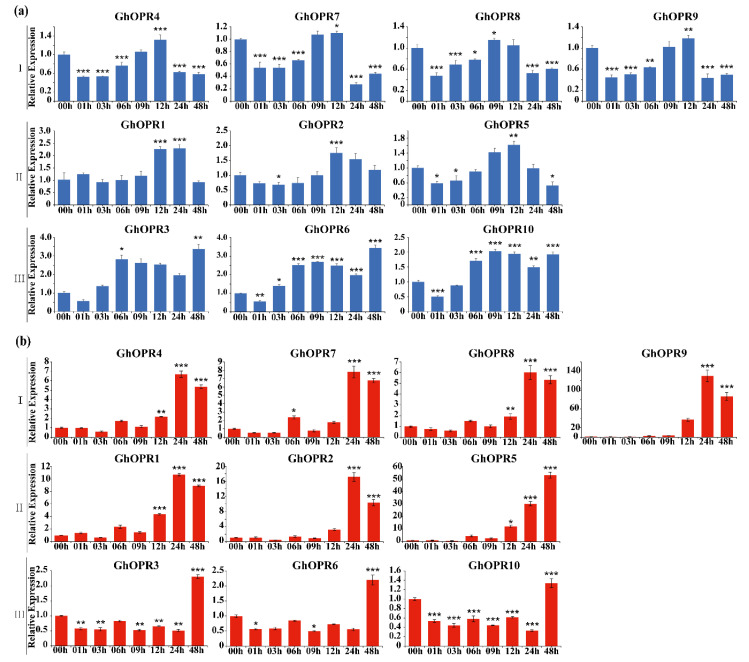
Analysis expression levels of *GhOPRs* under *V. dahliae* inoculation by RT-qPCR. The blue, red and black colors represent the sample of leaves, roots and stems, respectively. (**a**) The expression levels of *GhOPRs* under *V. dahliae* inoculation in leaf. (**b**) The expression levels of *GhOPRs* under *V. dahliae* inoculation in root. (**c**) The expression levels of *GhOPRs* under *V. dahliae* inoculation in stem. The samples were collected at 0, 1, 3, 6, 9, 12, 24, and 48 h after Vd080 inoculation. Values represent means ± standard deviation of three replicates. Asterisks reveal the gene significantly higher or lower in other time points than in 0 h by *t*-test (* *p* < 0.05, ** *p* < 0.01, *** *p* < 0.001).

**Figure 6 genes-11-01134-f006:**
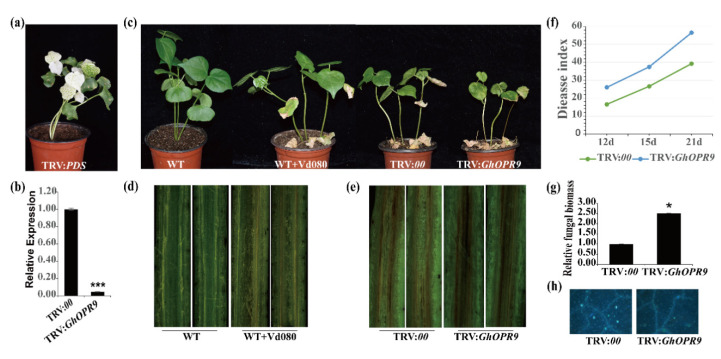
*GhOPR9* positively regulates cotton resistance against *V. dahliae* in the resistant *G. hirsutum* cv. Zhongzhimian No.2. (**a**) TRV:*PDS* was used as the indicator to evaluate VIGS. (**b**) The expression level of *GhOPR9* in the TRV:*00* and TRV:*GhOPR9* plants. Total RNA was isolated from roots at 10 days post-agroinfiltration. *GhUB7* was used as the reference. Each experiment was performed using three independent replicates. (**c**) Disease symptoms of the cotton plants after Vd080 infection. Photographs were taken at 21 days after inoculation. (**d**) Disease symptoms in stems of the WT plants at 21 days after Vd080 inoculation. Vascular browning was appeared in WT + Vd080 plants. (**e**) Disease symptoms in stems of the TRV:*00* and TRV:*GhOPR9* plants. (**f**) Disease index of the TRV:*00* and TRV:*GhOPR9* plants at 12 days, 15 days, and 21 days after inoculation with Vd080. Each experiment was performed using three replicates. (**g**) qPCR analysis of the relative fungal biomass in stems of the TRV:*00* and TRV:*GhOPR9* plants at 21 days after Vd080 inoculation. Each experiment was performed using three replicates. Differences between groups were compared using the *t*-test (* *p* < 0.05, *** *p* < 0.001). (**h**) Callose deposition in leaves of the TRV:*00* and TRV:*GhOPR9* plants at 21 days after Vd080 inoculation. Leaves were imaged on fluorescence microscopy.

**Figure 7 genes-11-01134-f007:**
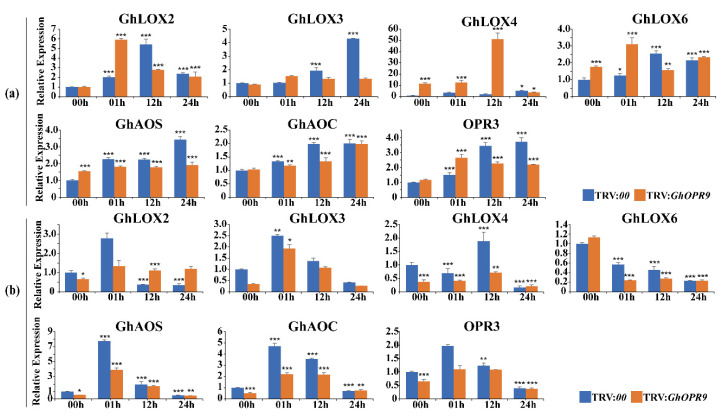
Real-time quantitative PCR analysis of the expression of JAs biosynthesis pathway genes at 00, 01, 12 and 24 h after Vd080 inoculation in the TRV:*00* and TRV:*GhOPR9* plants. (**a**) The expression levels of JAs biosynthesis pathway genes under Vd080 inoculation in leaf. (**b**) The expression levels of JAs biosynthesis pathway genes under Vd080 inoculation in root. Values represent means ± standard deviation of three replicates. The TRV:*00* plants 00 h were used as the control. Asterisks reveal the gene significantly higher or lower in other time points than in 00 h by *t*-test (* *p* < 0.05, ** *p* < 0.01, *** *p* < 0.001).

**Table 1 genes-11-01134-t001:** Prediction of cis-acting regulatory elements about various responses of *GhOPR* genes.

Element of Response	GhOPR1	GhOPR2	GhOPR3	GhOPR4	GhOPR5	GhOPR6	GhOPR7	GhOPR8	GhOPR9	GhOPR10
ABA	3		1		1	1	1	2		
anaerobic		1				1		1	1	3
auxin		1		2	1		1		1	
cold		1	3			1		3		
defense and stress	1		1	1		1				2
dehydration				3	2	1	1	4	1	2
drought					1		1		1	1
ETH	5	1	2	5		4	2	10	6	
GA		2	1		11		1		2	
osmotic stress, nutrient starvation	7	3	8	5		8	3	3	4	
JA									1	
light	15	3	9	9	8	6	11	4	4	8
MeJA				6		2	2	6	2	
SA		1		1	5		1			
stress	1			1					1	1
wounding and pathogen	2	3		2	2		1		6	2
wounding	15	25	19	21	16	17	23	10	22	27

The prediction of cis-acting regulatory elements were identified via the online database PlantCARE by using the 1.5 kb upstream of the allotetraploid cotton *G. hirsutum OPR* genes. The table list the number of cis-acting elements to various responses.

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
