# Peer review of "Genome-Wide Analysis of OPR Family Genes in Cotton Identified a Role for GhOPR9 in Verticillium dahliae Resistance"

_genes, 2020, doi:10.3390/genes11101134_

Round 1
Reviewer 1 Report
I have read the manuscript entitled “GhOPR9 modulates the resistance of cotton to Verticillium dahliae through regulating the expression of JAs pathway genes”, by Liu et al. The manuscript is well presented, while there are some concerns with respect to the manuscript should be resolved. Please, make sure that genes and scientific plant names are written in italics. Line 75: Please, replace showd by showed. Line 131: Plants were grown… Line 131: It should say growth chamber. Section 2.6.: How old were plants at the time of inoculation? Line 162: The fungus was grown … Lin 181: numbers of K2POH4 should be written as subscript. Lines 198: OPR Gene Family. Line 218: which was consistent with previous studies… Line 235: colors. Line 326: Element of response. Line 334: Please, check whether GhOPRs were induced by both abiotic stresses, I think the induction is only observed under NaCl stress. PEG6000 represses the relative expression of almost all the studied genes. Line 340: Please check this line, I think it should be “GhOPR3, GhOPR6 and GhOPR10 got the peak expression levels at 12 h”. Figure 4: Figure quality should be improved. Figure 5: Figure quality should be improved. Section 3.9: Why did you choose this time point (24 hours post infection) to analyze the relative expression of JA-regulated defence genes?Author Response
Please see the attachment.

Reviewer 2 Report
The authors in their manuscript present a phylogenetic characterization of OPRs in Gossypium species, a molecular (gene expression) study of G. hirsutum OPRs and functional characterization of a selected GhOPR using a gene silencing assay. The scientific concept approach and the methods selected are correct, however, the methodological approach concerning the functional characterization and the corresponding gene expression assays, as (applied) presented in the Materials & Methods and the Results sections, questions the validity of the results presented and the conclusions deduced. Furthermore, the manuscript should definitely be revised by an English language-native speaker/scientist, since there are a lot of grammar/syntax, expression and spelling errors, as well as scientific terminology editing and general text-editing mistakes, throughout the whole manuscript.
In specific:
Regarding the scientific part,
- The title is somehow misleading since it is based on the functional characterization of GhOPR9 and the results as presented and discussed do not clearly support this statement (see relevant comment No13). The authors are encouraged to reconsider a more representative title or keep the current one once the questions regarding the VIGS assay and the relative gene expression study are answered.
- In the M&M section 2.6, the authors should describe in more detail the reason (resistances, other characteristics, etc) for selecting the specific G. hirsutum cultivar.
- In the M&M section 2.7, the authors should clarify the difference between “replicate” and “biological replicate” (lines 139-140). It is not clear what is considered as a biological replicate and how many plants (or biological replicates?) form a sample.
- In the M&M sections 2.8 and 2.9, the authors should describe in detail the experimental design regarding all type of controls and treated plants used (infected/not infected, mock/empty vector/construct(s), etc..).
- In the Results section 3.1 there are specific references used next to the plant species named (lines 204-207). Are these references relevant for a reason to the specific genomic analysis performed? These references are not for the corresponding plant genomic databases, are references concerning similar scientific works and should be used in the relevant discussion section.
- In the Results section 3.2 the scope of the analysis is to test the ORP phylogenetic relationship, however, the ending conclusion refers to something very specific regarding OPR3 and JA, giving the impression of an “unfocused” aim of performing this type of analysis. An ending phrase regarding a more generalized conclusion reflecting the phylogenetic relationship would be possibly more appropriate.
- In the Results section 3.3 the authors should explain the role of these motifs in relation to the functions of the proteins.
- In the Results section 3.4 the supplementary material provided by the authors does not contain Figures. Thus, Figure S1 (and all others) cannot be accessed and reviewed.
- In the Results section 3.5 the authors refer to the specific analysis as of “Gossypium” (line 305). However, there are only two species of which the OPR promoter regions are analyzed and presented in the manuscript. In addition, since only the G. hirsutum OPRs are used for all subsequent analyses (gene expression, VIGS) it is not clear why the authors perform an in silico promoter analysis for G. barbadense and furthermore, select to present the relevant data (for G. barbadense) as supplementary. The authors should consider either to perform more experiments regarding gene expression and VIGS for the other species as well, or retract the specific G. barbadense promoter analysis data since they are not related to the rest of the work that follows.
- In the Results section 3.5 the work results presented are reflecting a prediction of putative cis-elements. This in silico study is just a recording of putatively functional cis-elements. The fact that their sequences are present does not mean that are functional or that a trans factor binds on them. This fact conditions the way of presenting the putative cis-elements in the text and tables. To be more precise I comment on the following example: the Table 1 title “Number of cis-acting regulatory elements about various responses of GhOPR genes” states that these are “regulatory elements” and furthermore that are involved in “various responses”. However, the in-silico promoter analysis results are just “sequences” that “could possibly be functional cis-elements”. This could be shown for example using either, a promoter deletion analysis experiment, an in vivo-footprinting or a gel retardation assay, that would accompany the in-silico analysis results. For the moment, they remain predicted cis-elements. Even when characterized as functional-cis elements their involvement in the specific plant responses (e.g ABA, auxin, SA, stress etc) should be shown experimentally. Till then remain sequences for predicted cis-elements. Thus, the authors are encouraged to reform the presentation of these results and the discussion section as well.
- In the Results sections 3.6 and 3.7, the letters/indications of the graphs and axes are to small and cannot be read. The whole figure size should be reformed.
- In the Results section 3.7 the authors present in the same graph the results concerning 3 different tissues. It is not clear which sample is considered as “reference” with relative expression as “1”, to which they compare all others. Is it the same for all tissues or different for each one of them? They should also present this/these in the graph as in Figure 4.
- In the Results section 3.8 the authors should present all relevant controls and data. They should present the same series of transformed plants (control, mock transformed, plain vector, vector for efficiency, and construct) for non-infected and infected plants. Not all the data are presented in figure 6. Though VIGS seems efficient (figure 6b) it is not clear if the phenotype itself in some plants of figure 6c is due to the construct TRV:GhOPR9 or due to the empty vector. Control (non-transformed infected and not-infected plants) are missing from the image thus any conclusion cannot be deduced. The same applies for the data of figure 6e. The Disease indexes for all cases should be presented. This is my main argument, which for the moment and in the way the results are presented and discussed, is the one that could arrest the publication of the manuscript in its present form.
In addition, the letters/indications of the graphs and axes are to small and cannot be read. The whole figure size should be reformed.
- The same concept as for comment No13 applies for the Results section 3.9. The results present the relative expression only of the transformed plants. For example if the “reference” sample for comparison would be the untransformed-uninfected plant the scaling for all comparisons even between the TRV:00 and TRV:GhOPR9 could be different. Not including these controls in the assay could result to bias.
- Considering comments 1 to 14 the whole Discussion section should be adapted (reformed) accordingly.
Regarding the presentation part:
There are basic English grammar/syntax, expression and spelling errors such as: incorrect or mixed use of different tenses (present/past), single/plural mistakes and basically, expression issues throughout the whole text. As examples I refer to some of them:
Lines 23-26 and 32: the meanings are somehow simplified
Line 41: “very complicated” instead of “complex” or “dense”
Lines 36-80: use of different or not proper tenses
Lines 77-80: The meaning is not clear
Lines 81-82: “not well understood” instead of e.g. “not studied explicitly”
Lines 184-196 and 226-228: Not clear expression, grammar mistakes
Furthermore, correct use of scientific terms is a prerequisite. For example, it is “RT-qPCR” and nor “qRT-PCR”, in line 242 it is possibly “group” instead of “subfamily” a term with a specific meaning in phylogeny, editing errors such as use of both “normal case” and “italics” for the species description, use of capital letters for “V. Dahliae”, etc.
Editing errors such as these in line 6, 198, etc and not providing all supplementary material.
Round 2
Reviewer 2 Report
The authors in their revised manuscript replied to all the reviewer comments and proceeded to relevant modifications, amendments, and corrections in the text and supplementary material, which improved the presentation of the scientific concepted discussed, its argumentation, as well as the text presentation regarding English language and editing issues.
However, for the manuscript to be published the authors should address the following aspects:
[1]
In the Results section 3.8 and the relevant part in the discussion section, the authors do not fully cover the question regarding the issues raised previously in the reviewer’s comments No 13 and 14, which remains blur.
To be specific:
for the VIGS experiment the authors should have presented in figure 6c and 6e (merged in only one photo) all relevant controls (as stated previously in the first revision).
In detail:
irrespective the control experiment with the TRV:PDS plants, which serves as a control defining the time to start the infection with V. dahliae (presented in a separate photo), in the putative-unified 6c and 6e photo the following plants/constructs and treatments should have been presented: a) WT plants, b) TRV:00, c)TRV:GhOPR9, d) WT plants infected with V. dahliae, e) TRV:00 infected with V. dahliae, f)TRV:GhOPR9 infected with V dahliae.
This is a standard experiment which does not miss any controls and a visual comparison between all cases would clarify any question.
The reason for this is that it is not clear (nor in the photos presented nor by any other experiment results), if: (i) the TRV:00 does not cause per se a or any phenotype when plants are transformed, and (ii) that does not cause a/any phenotype per se when plants are transformed and are also infected with Vd (synergistic action).
In the authors manuscript, if the vector alone does not cause or causes a phenotype to the plants, cannot be said (since these controls are missing..), and only assumptions that the VIGS system-vector does not affect the plants phenotype in any way, can be made.
The important thing it is not “if it causes or does not cause a phenotype per se”, but the integrity of the experiment and a full argumentation on that to be made/shown/written in the manuscript.
The authors defended the questions raised in the previous comments No 13 & 14 providing (among their reasoning) references No 27, 40, 61, 63, 65 which are publications with similar approaches regarding VIGS. Interestingly, nor in these publications the relevant controls are presented in their photos.
I searched the bibliography and reached the references of Gao et al., and Chua et al., ---(Gao X, Wheeler T, Li Z, Kenerley CM, He P, Shan L (2011a) Silencing GhNDR1 and GhMKK2 compromises cotton resistance to Verticillium wilt. Plant J 66: 293-305; Gao X, Britt RC, Jr., Shan L, He P (2011b) Agrobacterium-mediated virus-induced gene silencing assay in cotton. J Vis Exp 54: e2938; Jing Qu, Jian Ye, Yun-Feng Geng, Yan-Wei Sun, Shi-Qiang Gao, Bi-Pei Zhang, Wen Chen, and Nam-Hai Chua, Dissecting Functions of KATANIN and WRINKLED1 in Cotton Fiber Development by Virus-Induced Gene Silencing Plant Physiol. 2012 Oct; 160(2): 738–748.), --- in order to verify that the above issue is resolved in their research works when setting up the VIGS assay for cotton and Verticillium dahliae, however, the question of either TRV:00 has or does not have per se a phenotypic effect in cotton plants is not reported, nor discussed either in their texts…
I presume that since there was any effect observed, the authors (Gao, Chua and all other authors in subsequent publications) base their results on the “assumption” that the VIGS TRV:00 does not cause any phenotype in cotton when used.
However, it is appropriate to mention that in numerous other scientific publications where the VIGS system is used (regarding fungal pathogens and viruses), all controls are presented, or at least discussed.
It is also clearly demonstrated that the VIGS system and TRV:GhOPR9 is functional in the present experiment as this is deduced from the results and photos 6b, 6g, 6h and 6i. However, a clear difference between TRV:00 and TRV:GhOPR9 cannot be easily deduced from the photo 6e.
Concerning all the above reasons/facts, I am not in the position to ask from the authors to present a photo with all the controls needed/described.
Thus,
- if the authors do have such a photo/photos it would be better to present it instead of 6c and 6e.
- they should also incorporate, somewhere in their text regarding the VIGS system presentation or relevant, the references of Gao et al., and Chua et al., mentioned above.
- They should present figures 6d and 6f with the same background “lighting”. 6f is much darker and comparisons cannot be made easily.
[2]
The authors proceeded to a manuscript revision regarding English language by using external help for revision. The effort is highly appreciated and there is improvement regarding editing issues, however, there are numerous points where grammar/syntax, expression and spelling errors remain through the whole manuscript.
In specific, such points are found/tracked in the following lines of the revised manuscript:
23, 30, 37 (twice), 46, 47, 50, 51 (twice), 52, 68, 75, 76, 77, 78, 80, 86 to 90 (meaning), 91, 92, 94, 95, 97, 98, 105, 110, 119, 133, 135, 136, 148, 166, 189, 190, 194, 195, 224, 225, 251 (expression), 254(incomplete sentence), 258, 260, 264, 274, 297, 298, 302, 304, 318 (twice), 330, 333, 339, 354, 355, 363, 378 to 380 (meaning), 392, 403, 405, 407, 409, 411, 416, 429 (infection instead inoculation), 441, 442, 444, 447, 455 (twice), 466 to 473 (expression), 477 (twice), 480, 481, 482, 483, 485, 494, 502, 504, 517 to 522 (meaning), 530, 533, 534, 535, 543, 544 and 545 (meaning), 558, 566, 572, 573, 576, 590, 592.
The authors should ask for revision of the manuscript by an English language-native speaker/scientist or a relevant journal editing service before publication.
